# Ensartinib (X-396) Effectively Modulates Pharmacokinetic Resistance Mediated by ABCB1 and ABCG2 Drug Efflux Transporters and CYP3A4 Biotransformation Enzyme

**DOI:** 10.3390/cancers12040813

**Published:** 2020-03-28

**Authors:** Dimitrios Vagiannis, Eva Novotna, Adam Skarka, Sarah Kammerer, Jan-Heiner Küpper, Si Chen, Lei Guo, Frantisek Staud, Jakub Hofman

**Affiliations:** 1Department of Pharmacology and Toxicology, Faculty of Pharmacy in Hradec Kralove, Charles University, Heyrovskeho 1203, 500 05 Hradec Kralove, Czech Republic; vagiannd@faf.cuni.cz (D.V.); staud@faf.cuni.cz (F.S.); 2Department of Biochemical Sciences, Faculty of Pharmacy in Hradec Kralove, Charles University, Heyrovskeho 1203, 500 05 Hradec Kralove, Czech Republic; novotne7@faf.cuni.cz; 3Department of Chemistry, Faculty of Science, University of Hradec Králové, Hradecká 1285, 500 03 Hradec Králové, Czech Republic; adam.skarka@uhk.cz; 4Institute of Biotechnology, Brandenburg University of Technology Cottbus-Senftenberg, Universitätsplatz 1, 01968 Senftenberg, Germany; sarah.kammerer@b-tu.de (S.K.); jan-heiner.kuepper@b-tu.de (J.-H.K.); 5Division of Biochemical Toxicology, National Center for Toxicological Research/U.S. FDA, 3900 NCTR Road, Jefferson, AR 72079, USA; Si.Chen@fda.hhs.gov (S.C.); Lei.Guo@fda.hhs.gov (L.G.)

**Keywords:** ensartinib, cancer, multidrug resistance, drug-drug interaction, ABC transporter, cytochrome P450

## Abstract

Ensartinib (X-396) is a promising tyrosine kinase inhibitor currently undergoing advanced clinical evaluation for the treatment of non-small cell lung cancer. In this work, we investigate possible interactions of this promising drug candidate with ATP-binding cassette (ABC) drug efflux transporters and cytochrome P450 biotransformation enzymes (CYPs), which play major roles in multidrug resistance (MDR) and pharmacokinetic drug-drug interactions (DDIs). Accumulation studies showed that ensartinib is a potent inhibitor of ABCB1 and ABCG2 transporters. Additionally, incubation experiments with recombinant CYPs showed that ensartinib significantly inhibits CYP3A4 and CYP2C9. Subsequent molecular docking studies confirmed these findings. Drug combination experiments demonstrated that ensartinib synergistically potentiates the antiproliferative effects of daunorubicin, mitoxantrone, and docetaxel in ABCB1, ABCG2, and CYP3A4-overexpressing cellular models, respectively. Advantageously, ensartinib’s antitumor efficiency was not compromised by the presence of MDR-associated ABC transporters, although it acted as a substrate of ABCB1 in Madin-Darby Canine Kidney II (MDCKII) monolayer transport assays. Finally, we demonstrated that ensartinib had no significant effect on the mRNA-level expression of examined transporters and enzymes in physiological and lung tumor cellular models. In conclusion, ensartinib may perpetrate clinically relevant pharmacokinetic DDIs and modulate ABCB1-, ABCG2-, and CYP3A4-mediated MDR. The in vitro findings presented here will provide a valuable foundation for future in vivo investigations.

## 1. Introduction

Lung cancer is a major cause of cancer deaths in both genders, with the non-small cell lung cancer (NSCLC) subtype accounting for almost 85% of all lung cancer cases [1]. It has traditionally been treated with non-selective cytotoxic drugs that cause severe harm to physiological tissues. These agents are gradually being replaced by novel targeted therapies that selectively affect cancer-specific regulatory pathways, dramatically improving the safety of anticancer treatment [2]. At present, the most clinically successful group of targeted drugs are tyrosine kinase inhibitors (TKIs) and therapeutic antibodies. Ensartinib (X-396) is a third-generation TKI (Figure 1) that is a potent inhibitor of mutated anaplastic lymphoma kinase (ALK); deregulation of this enzyme has been found to be a meaningful target in 3–7% of all NSCLC cases [3]. Ensartinib is currently undergoing phase II and III clinical trials for the treatment of NSCLC [4]; preclinical investigations have shown that it is more active than the approved ALK inhibitors crizotinib, alectinib, and ceritinib and that it retains activity even in models with ALK mutations that confer resistance to these agents [5].

During treatment, many patients develop drug resistance and initially sensitive tumor cells stop responding to the applied drugs. This is a major therapeutic obstacle that has been reported for both classical cytotoxic and targeted drugs, and can arise via pharmacodynamic and/or pharmacokinetic mechanisms [6]. Common mechanisms of pharmacodynamic resistance include overexpression or mutation of the drug’s target, whereas pharmacokinetic resistance occurs primarily as a result of enhanced drug efflux or drug deactivation by metabolic enzymes [7].

Drug efflux can be mediated by ATP-binding cassette (ABC) transporters, which collectively constitute one of the biggest membrane transport protein superfamilies. Several members of this superfamily are predominantly localized in normal tissues of diverse organs [8]. ABC drug efflux transporters can pump drugs and other xenobiotics out of the cells and thereby protect sensitive tissues against the harmful effects of these compounds. However, some of these proteins are also overexpressed in certain tumor cells, where they contribute to the emergence of multidrug resistance (MDR). Dozens of ABC transporters have been described, but only three have unambiguously been shown to play a role in MDR in vitro and in vivo: ABCB1 (P-glycoprotein), ABCG2 (breast cancer resistance protein), and ABCC1 (MDR-associated protein 1) [7,9]. In addition to their role in MDR, ABCB1 and ABCG2 are recognized mediators of potentially dangerous and clinically relevant pharmacokinetic drug-drug interactions (DDIs) [10]. Consequently, multiple drug regulatory authorities have highlighted the clinical importance of identifying interactions of novel drugs with ABC transporters [11,12].

In addition to drug transport, pharmacokinetic resistance can emerge due to the activity of drug metabolizing enzymes. Most metabolic reactions, which greatly increase the likelihood of drug elimination, are mediated by a handful of cytochrome P450 (CYP) isoforms—notably, CYP3A4, CYP3A5, CYP2D6, CYP2C8, CYP2C9, CYP1A2, CYP2B6, and CYP2C19. Like ABC transporters, these enzymes can act as important perpetrators of pharmacokinetic DDIs in a patient’s body [13]. CYPs promote the metabolic conversion of various cytotoxic agents, and the pharmacodynamic activities of the resulting degradation products generally differ from that of the parent drug. For example, the detoxification of cytotoxic agents such as paclitaxel, docetaxel, and vincristine has been linked to the activities of CYP3A4/5 and CYP2C8 isoforms in cancer cells [14,15,16]. Significant intratumoral expression of CYPs has been observed in several cancer types [17], offering a seemingly straightforward explanation for these outcomes. However, unpublished studies conducted in our group indicate that only CYP3A4 significantly attenuates the cytotoxic activity of docetaxel in experimental models.

Over the last three decades, several clinical trials evaluating combinations of non-cytotoxic ABC transporter inhibitors with MDR-susceptible cytotoxic drugs have been conducted to identify therapeutic regimens able to circumvent transporter-mediated resistance. Despite initially promising results, these combinations have failed to meet expectations due to inadequate efficacy and/or high toxicity [18]. However, several new targeted drugs act as MDR modulators with dual mechanisms of action, exhibiting both intrinsic antiproliferative activity and an ability to inhibit ABC transporters. This dual activity has attracted considerable interest, and combinations of these new drugs with standard cytostatic agents may be powerful therapeutic options for resistant tumors [19,20]. Our research group recently demonstrated that several cyclin-dependent kinase inhibitors exhibit such dual modes of action [21,22,23,24,25,26,27]. We believe that the synergistic combinations identified in our studies could potentially be translated into more efficient and safe therapies for many oncological patients.

In this work, we investigated the interactions of ensartinib with ABC drug efflux transporters and CYPs, and the effects of these interactions on MDR. Both in vitro and in silico methods were used, including cell accumulation and transport studies, molecular docking experiments, antiproliferative combination assays, incubations with recombinant enzymes, and gene induction studies.

## 2. Results

### 2.1. Ensartinib Potently Inhibits ABCB1 and ABCG2 Transporters in MDCKII Sublines

We first evaluated the possible effect of ensartinib on the accumulation of hoechst 33342 and calcein AM in Madin-Darby Canine Kidney II (MDCKII) cell lines and then performed flow cytometric studies using the cytostatic drugs daunorubicin and mitoxantrone to confirm the initially obtained results.

Hoechst 33342 and calcein AM accumulation assays in MDCKII cells transduced with human ABC transporters showed that ensartinib significantly inhibited both ABCB1 (IC_50_ = 3.69 µM; Figure 2A) and ABCG2 (IC_50_ = 9.13 µM; Figure 2B), matching the potency of the model inhibitors LY335979 and Ko143, respectively. However, it was a much less potent inhibitor of ABCC1-mediated calcein AM efflux (IC_50_ > 50 µM; Figure 2C). Similar results were obtained in follow-up experiments using daunorubicin and mitoxantrone: ensartinib strongly increased the accumulation of these cytostatic agents in both MDCKII-ABCB1 (IC_50_ = 4.75 µM, Figure 2D) and MDCKII-ABCG2 (IC_50_ = 28.3 µM, Figure 2E) cells. In keeping with the results of the calcein AM experiments, we observed insignificant inhibition of daunorubicin efflux in MDCKII-ABCC1 cells (IC_50_ > 50 µM; Figure 2F). Ensartinib also had no significant effect on the accumulation of any substrate in control MDCKII-par cells (Figure 2A–F).

### 2.2. Ensartinib Blocks the Activity of Several Clinically Important CYP Isoforms

The inhibitory effect of ensartinib towards eight clinically relevant CYP isoforms was assessed to estimate its potential to cause pharmacokinetic DDIs and modulate CYP3A4-mediated docetaxel resistance. Ensartinib strongly inhibited the CYP3A4 and CYP2C9 isoforms with IC_50_ values of 1.12 and 4.93 µM, respectively. It also moderately inhibited CYP3A5 (IC_50_ = 19.1 µM), CYP2C8 (IC_50_ = 20.6 µM), and CYP2C19 (IC_50_ = 14.6 µM) but had no appreciable inhibitory effect on the CYP1A2, CYP2D6, and CYP2B6 isoforms (Figure 3).

### 2.3. Inhibition of CYP3A4 in Intact HepG2-CYP3A4 Cells by Ensartinib

These assays were performed to confirm the ability of ensartinib to modulate CYP3A4-mediated docetaxel resistance and perpetrate pharmacokinetic DDIs on CYP3A4. In contrast to the strong inhibitory potency observed towards the recombinant CYP3A4 isoenzyme, ensartinib caused only moderate inhibition of CYP3A4 in enzyme-overexpressing HepG2 cells (IC_50_ = 23.5 µM). The model inhibitor 10 µM ketoconazole induced 97.4% inhibition, confirming the validity of the applied method (Figure 4).

### 2.4. Molecular Docking of Ensartinib into ABCB1, ABCG2 and CYP3A4

We performed molecular docking simulations to characterize the molecular background of the interactions between ensartinib and ABCB1, ABCG2, and CYP3A4. Flexible molecular docking was performed into all three ligand binding sites of the inward-facing form of the ABCB1 transporter: the modulator site (M-site), a rhodamine-binding site (R-site), and a hoechst site (H-site) [28]. Ensartinib is best fitted to the M-site (−12.6 kcal/mol), where it is surrounded by protein residues Leu-65, Phe-336, Ile-340, Phe-343, Gln-347, Gln-725, Phe-728, Ala-729, Phe-732, Ser-979, Phe-983, Met-986, and Gln-990; it binds to the R- and H-sites with lower affinities (-9.3 and −9.5 kcal/mol, respectively) (Figure 5A). Ensartinib is also known to interact with the ATP-binding pocket of anaplastic lymphoma kinase (ALK) [29]. Therefore, to test the possibility that ensartinib might interfere with the ATP-binding pocket of ABCB1’s nucleotide binding domains (NBDs), we also performed flexible docking into this site. Docking results indicated that ensartinib binds to the ATP-binding pocket with an energy of −10.6 kcal/mol and may thus compete with ATP and thereby influence ABCB1 efflux activity (Figure 5B). This conclusion was further supported by the results of ABCB1 ATPase assay as ensartinib significantly decreased verapamil-stimulated ATPase activity of ABCB1. In addition, ensartinib also significantly increased baseline vanadate-sensitive ATPase activity, which suggest that it could be a substrate of ABCB1 (Figure 5C).

To describe the molecular background of ensartinib’s interactions with ABCG2, a flexible molecular docking study was performed using the dimeric form of this transporter. Ensartinib was predicted to bind to the internal cavity with high affinity (−12.9 kcal/mol), and one molecule of the ligand appeared to be sufficient to lock ABCG2 in its inward-facing conformation (Figure 5D). Further docking into the NBDs of ABCG2 revealed no conformation with an energy below −8.0 kcal/mol, suggesting that interference with ATP binding makes no appreciable contribution to the observed inhibitory effects towards ABCG2.

A flexible molecular docking study was also performed to study the binding interactions of ensartinib with CYP3A4. The crystal structure of CYP3A4 containing ketoconazole was downloaded, ketoconazole was removed, and ensartinib was docked into its position, revealing that it has a high predicted affinity for CYP3A4 (−10.6 kcal/mol) (Figure 5E).

### 2.5. Modulatory Effects of Ensartinib on ABC Transporter-Mediated Cytostatic MDR

We performed additional experiments to determine whether ensartinib’s inhibitory interactions with transporters could be exploited to circumvent resistance to the MDR-susceptible agents daunorubicin (ABCB1 and ABCC1) and mitoxantrone (ABCG2). This hypothesis was first tested in MDCKII cells and then in more clinically relevant models—human cytostatic resistant HL60 sublines. The modulatory concentration of ensartinib (10 µM) was chosen because (1) it induced appreciable ABC transporter inhibition, (2) it is potentially clinically relevant given the Cmax of 1 μM [30] determined in in vivo pharmacokinetic studies in humans, and (3) it has negligible cytotoxic effects in the tested cell lines.

In our drug combination studies, ensartinib completely sensitized cells overexpressing ABCB1 (R_R_ = 11.2) and ABCG2 (R_R_ = 6.16) to daunorubicin and mitoxantrone, respectively (Figure 6A,B, Table 1). Complete resistance reversal was also observed in HL60-ABCG2 cells (Figure 6D), with an R_R_ of 18.6 (Figure 6D, Table 1). Additionally, ensartinib effectively modulated the sensitivity of HL60-ABCB1 cells to daunorubicin (R_R_ = 22.1) despite not quite restoring them to the sensitivity of the parent cells (Figure 6C, Table 1). No significant IC_50_ shifts were observed in the parent variants of either cellular model. The R_R_ values determined for the transporter-overexpressing and parent cells suggest that transporter inhibition plays a dominant role in the observed modulation of cytostatic resistance. To confirm this conclusion, we applied the Chou-Talalay combination index method, which uses a mathematical algorithm to quantify the effects of drug combinations. The combination effects observed in the parental MDCKII and HL60 cell lines were antagonistic or additive across the whole FA range. Conversely, the CI values for their ABCB1- and ABCG2-overexpressing counterparts were in the synergistic range (Figure 7). These results clearly indicate that ensartinib-mediated transporter inhibition contributed substantially to the chemosensitization of the tested cells.

### 2.6. Ensartinib Attenuates CYP3A4-Mediated Docetaxel Resistance

Having confirmed that ensartinib inhibits CYP3A4 in cellular models, we evaluated its ability to overcome CYP3A4-mediated docetaxel resistance. A clinically relevant docetaxel concentration (1 µM) was used in these experiments [31]; unpublished studies from our group have shown that CYP3A4 contributes significantly to docetaxel resistance at this concentration. The tested ensartinib concentrations (15 and 25 µM) were chosen based on similar criteria to those applied when testing transporter-oriented drug combinations.

After treatment with 1 µM docetaxel, the viability of CYP3A4-overexpressing cells was 24.4% greater than that of cells transduced with the empty vector. This asymmetry was annulled by a non-cytotoxic 10 µM concentration of the model inhibitor ketoconazole, which had no significant effect on the antiproliferative activity of docetaxel in HepG2-EV cells (Figure 8A,B). These results clearly demonstrate that CYP3A4 contributes to docetaxel resistance and that its contribution can be reversed by a specific CYP3A4 inhibition. In CYP3A4-overexpressing cells, 25 µM (but not 15 µM) ensartinib caused docetaxel sensitization comparable to that induced by ketoconazole (Figure 8A). No such potentiation of docetaxel was observed in empty vector-transduced cells (Figure 8B). Ketoconazole and ensartinib are both slightly cytotoxic by themselves, therefore we compensated for this distorting influence by calculating sensitization effects (see Materials and Methods). Sensitization effect values may indicate antagonism (<0%), additivity (=0%), or synergism (>0%). As shown in Figure 8C, the combination of docetaxel with ketoconazole or ensartinib is naturally antagonistic (HepG2-EV data), but its effect becomes synergistic in CYP3A4-overexpressing cells. These results demonstrate that ensartinib can modulate docetaxel resistance by interacting with the CYP3A4 isozyme.

### 2.7. Ensartinib Is a Substrate of ABCB1, But Not of the ABCG2 and ABCC1 Transporters

We next sought to determine whether ensartinib is a transporter substrate and could therefore become MDR-susceptible. Cellular monolayer assays with MDCKII cells were performed to assess the affinity of ensartinib toward the tested transporters at a non-saturating concentration of 1 µM. The transport ratio (*r*) was calculated as the ratio of transport in the basolateral-to-apical (BA) direction to that in the apical-to-basolateral (AB) direction based on end-point data. According to FDA/EMA guidelines, a drug should be regarded as a transporter substrate if *r* ≥ 2 [11,12].

As expected, no asymmetry in bidirectional transport was observed in control MDCKII-par cells (*r* of 1.01; Figure 9A). Similar results were obtained with MDCKII-ABCG2 and MDCKII-ABCC1 cells, which exhibited an insignificant acceleration of ensartinib transport in the BA direction (*r* = 1.15 and *r* = 1.29; Figure 9C and 9D, respectively). However, ensartinib proved to be an ABCB1 substrate: an *r* value of 15.6 was observed in MDCKII-ABCB1 cells. This transport asymmetry was completely abolished by adding 1 µM LY335979 (*r* = 1.05), confirming that ensartinib is transported by ABCB1 (Figure 9B). Obtained data correlate well with those from ABCB1 ATPase activation experiments (Figure 5C).

### 2.8. ABCB1, ABCG2, and ABCC1 Transporters Do Not Confer Resistance to Ensartinib

We conducted follow-up studies to verify the results of the preceding transport studies and further explore the possible susceptibility of ensartinib to transporter-mediated MDR. Comparative proliferation experiments were therefore performed in MDCKII, HL60, and A431 cellular models with and without transporter overexpression.

In accordance with the results of the bidirectional transport assays, no significant differences were observed between MDCKII/HL60/A431-par cells (IC_50_ = 21.8/31.9/2.69 µM) and the corresponding ABCG2- (IC_50_ = 23.3/24.3/2.26 µM) or ABCC1-overexpressing (IC_50_ = 21.9/27.4/1.86 µM) cells. Interestingly, although ensartinib was characterized as an ABCB1 substrate in transport experiments, it was not more effective in MDCKII/HL60/A431-par cells than in their ABCB1-overexpressing counterparts (IC_50_ = 23.9/29.6/2.79 µM) (Figure 10). Taken together, these results suggest that functional overexpression of ABCB1, ABCG2, and ABCC1 does not confer resistance to ensartinib.

### 2.9. Changes in ABCB1, ABCG2, ABCC1, and CYP1A2, CYP3A4, CYP2B6 Expression Following Exposure to Ensartinib

Pharmacokinetic DDIs could also result from changes in the gene expression of ABC transporters and CYP enzymes following ensartinib treatment. We therefore investigated the effect of ensartinib on the mRNA levels of *ABCB1*, *ABCG2*, *ABCC1*, and *CYP1A2*, *CYP3A4*, *CYP2B6* in intestine (LS174T and Caco-2 cells) and liver (HepaFH3 cells and upcyte hepatocytes) models. Additionally, mRNA gene induction studies were performed in NSCLC cellular models (A549 and NCI-H1299) to determine whether ensartinib could influence the MDR phenotype of its main target cancer cells.

The ensartinib concentration used in these experiments (0.5 µM) was chosen because (1) it exhibits negligible cytotoxicity in the tested cell lines (Figure 11A, Figure 12A), and (2) Cmax values observed during in vivo pharmacokinetic studies in patients [30] suggest that it is a pharmacologically relevant concentration. The tested drug did not increase or reduce the mRNA levels of *ABCB1*, *ABCG2*, or *ABCC1* by more than 100% or 50% in any case (Figure 11B–E). Similar results were observed in experiments focusing on the CYP isoforms *CYP1A2*, *CYP3A4*, and *CYP2B6* (Figure 12B–E). According to the EMA guidelines [11], we can conclude that ensartinib has no potential to perpetrate induction-based DDIs or to influence the MDR phenotype of tumor cells due to changes in gene expression.

## 3. Discussion

Ensartinib is a promising tyrosine kinase inhibitor currently undergoing advanced clinical evaluation for the treatment of ALK-positive NSCLC [29]. While its basic pharmacokinetic properties were characterized during its development [32], its possible interactions with drug efflux transporters and biotransformation enzymes have yet to be clarified. This study therefore explored the interactions of ensartinib with several ABC transporters and CYP isoenzymes to evaluate its potential usefulness for overcoming pharmacokinetic MDR.

Our study began with experiments on the inhibitory affinity of ensartinib for ABC transporters and CYP enzymes, which play key roles in pharmacokinetic DDIs and anticancer drug resistance. Accumulation studies revealed ensartinib to be a potent inhibitor of ABCB1 and ABCG2, but not of ABCC1. Interestingly, the inhibitory affinity of the tested TKI was almost identical for ABCG2 but varied by a factor of 3 for ABCB1 among model substrates used. Molecular docking analysis revealed a possible explanation for this phenomenon: ensartinib inhibits ABCG2 by interacting with a single ligand binding site, but interacts with all three binding sites of ABCB1 with different energies. Since model substrates also bind to these binding sites with varying strengths [28], the background of the interaction is very complex, which could explain the observed differences in inhibitory affinity. Similar variance was observed in previous studies [25,26], highlighting the importance of using multiple model substrates when investigating the potential interactions of transporters with multiple binding sites. In addition to the ABCB1 and ABCG2 transporters, ensartinib potently inhibited recombinant CYP3A4 and CYP2C9 with IC_50_ values below 5 µM. Although the CYP3A4 interaction was confirmed by a molecular docking simulation, the effect of ensartinib on this enzyme in cellular model systems was comparatively modest. This is unsurprising because the intracellular activities of ligands are known to be limited by factors such as their capacity for membrane penetration, their distribution and transport by efflux transporters or enzymes, trapping in endosomes, cytotoxicity, and so on [33]. Importantly, because ABCB1, ABCG2, CYP3A4, and CYP2C9 interacted with ensartinib at concentrations that may occur in the plasma of oncological patients (Cmax = 1 μM) [30], these transporters and enzymes are potential perpetrators of clinically relevant DDIs. Currently unavailable quantities such as the extent of plasma protein binding and the unbound fraction of Cmax for ensartinib would enable a more thorough assessment of their possible roles in this context [11,12].

After the inhibition experiments, we investigated the possibility of exploiting the observed interactions to attenuate pharmacokinetic chemoresistance. Ensartinib was found to effectively modulate ABCB1-, ABCG2-, and CYP3A4-mediated resistance to daunorubicin, mitoxantrone, and docetaxel, respectively, in a synergistic fashion. This is a very clinically relevant finding; synergistic effects of drug combinations are believed to make important contributions that are often concealed behind the clinical successes of various anticancer chemoregimens. Synergism enables drug dose reduction while maintaining adequate therapeutic efficacy, which can substantially improve the safety of treatments [34]. No in vitro or in vivo studies on the co-administration of ensartinib with other anticancer drugs have been reported previously. We thus provide the first evidence of possible therapeutic benefits arising from the combination of ensartinib with standard cytotoxic drugs. In addition to ensartinib, MDR-modulatory properties have been reported for other ALK-targeting drugs. First-generation ALK inhibitor crizotinib was shown to enhance the efficacy of doxorubicin and paclitaxel via ABCB1 inhibition in vitro and in vivo, respectively [35]. In addition, second-generation ALK inhibitors ceritinib and alectinib were both described as dual ABCB1 and ABCG2 inhibitors able to potentiate anticancer effects of chemotherapeutic drugs, which are substrates of these transporters [36,37]. Noteworthy, these properties are not limited only to ALK inhibitors but were observed also in TKIs hitting other targets, such as epidermal growth factor receptor (EGFR) inhibitor gefitinib, vascular endothelial growth factor receptor (VEGFR) inhibitor sorafenib and several others [19,20]. Currently, few MDR-antagonizing combinations including those with erlotinib, lapatinib, nintedanib, and sorafenib have being evaluated in clinical trials for the treatment of various cancer types [38].

Although we introduced ensartinib as an effective resistance modulator, it might also be susceptible to MDR itself. To test this possibility, we performed a series of experiments including monolayer transport studies and comparative viability assays in cells transduced with MDR-associated transporters. Ensartinib exhibited no significant substrate affinity for ABCG2 or ABCC1, in keeping with the finding that the presence of these transporters had no effect on its antiproliferative efficacy. Interestingly, although ABCB1-mediated ensartinib transport was demonstrated in bidirectional transport assays, functional expression of ABCB1 in three cellular models had no appreciable impact on ensartinib’s antitumor activity. Ensartinib is a relatively lipophilic drug (predicted logP ≈ 3.6 according to Advanced Chemistry Development Software version 11.02), whereas common MDR-susceptible drugs such as anthracyclines, mitoxantrone, and epipodophyllotoxins are predominantly hydrophilic, with logP values around 1. It is well-known that moderate rates of transport by efflux systems can be counteracted by passive diffusion in the case of lipophilic drugs [39], which may explain the observed discrepancy. Ensartinib’s efficacy is thus unaffected by functional transporter expression. Together with its targeting of multiple transporters, this makes it an ideal chemosensitizer [40].

Finally, we investigated the effects of ensartinib on the mRNA levels of clinically relevant ABC efflux transporters and CYP isoforms. Based on our data and EMA guidelines [11], we can conclude that ensartinib is unlikely to perpetrate induction-based DDIs or potentiate the resistance phenotype of NSCLC cells. The latter in particular strengthens the potential value of ensartinib as an MDR modulator. To our knowledge, no previously published studies have explored the possible influence of ensartinib on the expression of ABC drug efflux transporters and CYPs associated with drug resistance or DDIs.

## 4. Materials and Methods 

### 4.1. Reagents and Chemicals

Ensartinib was obtained from MedChem Express (New Jersey, NJ, USA). Hoechst 33342, daunorubicin, mitoxantrone, MTT, XTT, dimethyl sulfoxide (DMSO), fluorescein isothiocyanate-labeled dextran, phenazine methosulfate, and cell culture reagents as well as CYP inhibitors (α-naphthoflavone, miconazole, montelukast, sulfaphenazole, quinidine and ketoconazole) and CYP/transporter inducers (rifampicin, omeprazole and phenobarbital) were purchased from Sigma Aldrich (St. Louis, MO, USA). Calcein AM, Vivid CYP Screening Kits, and Pierce BCA protein Assay Kits were supplied by Thermo Fisher Scientific (Waltham, MA, USA). DNase I, Reaction Buffer with MgCl_2_, 50 mM EDTA, oligo (dT)18 primers, 10 mM dNTPs for DNase digestion, RevertAid reverse transcriptase for cDNA synthesis, and Maxima Probe qPCR Master Mix for the analysis of *CYP1A2*, *CYP2B6* and *CYP3A4* expression in HepaFH3 cells and upcyte hepatocytes were also obtained from Thermo Fisher Scientific (Waltham, MA, USA). The InnuPREP RNA Mini Kit was bought from Analytik Jena (Jena, Germany). EvaGreen was obtained from Biotium (Fremont, CA, USA). LY335979 (zosuquidar) was obtained from Toronto Research Chemicals (North York, ON, Canada). Ko143 and MK-571 were from Enzo Life Sciences (Farmingdale, NY, USA). Opti-MEM was bought from Gibco BRL Life Technologies (Rockville, MD, USA). Hepatocyte Culture Medium and Hepatocyte High Performance Medium were from Upcyte Technologies (Hamburg, Germany). TRI reagent was purchased from the Molecular Research Center (Cincinnati, OH, USA). TaqMan systems for the analysis of *ABCB1*, *ABCG2*, *ABCB1*, and *CYP3A4* mRNA expression in A549 and NCI-H1299 cells, gb Reverse Transcription Kit, and gb Easy PCR Master Mix were purchased from Generi Biotech (Hradec Kralove, Czech Republic). The P450-Glo CYP3A4 Assay and Screening System with Luciferin-IPA, the CellTiter-Glo Luminescent Cell Viability Assay kit, and the CellTiter-Glo 2.0 Cell Viability Assay kit were bought from Promega (Madison, WI, USA). ABCB1 PREDEASY ATPase kit (SB MDR1/P-gp) was purchased from Solvo Biotechnology (Szeged, Hungary). All other chemicals and reagents were of the highest purity commercially available.

### 4.2. Cell Culture

Parent Madin-Darby canine kidney II (MDCKII-par) cells and their ABC transporter-transduced counterparts (MDCKII-ABCB1, MDCKII-ABCG2, and MDCKII-ABCC1) were obtained from Dr. Alfred Schinkel (The Netherlands Cancer Institute, Amsterdam, Netherlands). Human squamous carcinoma A431-parent cells and their ABCB1-, ABCG2-, and ABCC1-overexpressing variants were provided by Dr. Balasz Sarkadi (Hungarian Academy of Sciences, Budapest, Hungary) [41]. Human leukemia cell line HL60 parent cells and ABCB1, ABCG2, and ABCC1 transporter-overexpressing HL60 cells were also supplied by Dr. Balazs Sarkadi (Hungarian Academy of Sciences, Budapest, Hungary) [42,43]. Human liver carcinoma HepG2 cells stably transduced with human CYP3A4 (HepG2-CYP3A4) together with the empty vector transduced (HepG2-EV) subline were generated as described previously [44]. Human NSCLC cell lines A549 and NCI-H1299 as well as the human colorectal adenocarcinoma cell line Caco-2 were obtained from the American Type Culture Collection (Manassas, VA, USA). The second human colorectal adenocarcinoma cell model LS174T was from the European Collection of Cell Cultures (Salisbury, UK). HepaFH3 proliferation-competent primary-like human hepatocytes were produced using upcyte technology and characterized as described previously [45]. Upcyte Technologies (Hamburg, Germany) prepared second generation upcyte hepatocytes, which were derived from another donor (no. 653-03) [46]. MDCKII, A431, HepG2, and A549 cells were cultivated in high-glucose Dulbecco’s modified Eagle’s medium (DMEM) supplemented with 10% fetal bovine serum (FBS). HL60 cells were propagated in Roswell Park Memorial Institute medium 1640 (RPMI-1640) with 10% FBS. Caco-2 cells were grown in high-glucose DMEM supplemented with 10% FBS and 1% non-essential amino acids. NCI-H1299 cells were cultured in RPMI-1640 supplemented with 10% FBS, 1 mM sodium pyruvate, and 10 mM HEPES. For LS174T cells, Eagle’s minimal essential medium (EMEM) supplemented with 2 mM glutamine, 1% non-essential amino acids, and 10% FBS was used. HepaFH3 and upcyte hepatocytes were routinely cultured in complete Hepatocyte Culture Medium, while complete Hepatocyte High-Performance Medium was used in experiments. All routine cultivations and experiments were conducted under antibiotic-free conditions with the exception of monolayer transport assays, in which the media contained 1% penicillin-streptomycin. Cells used in experiments were from passages 10 to 25 and were periodically tested for mycoplasma infection. The concentration of dimethyl sulfoxide (DMSO), which was used as a solvent for ensartinib, did not exceed 0.5% in experiments; possible distortions of results due to the use of DMSO were avoided by using vehicle controls.

### 4.3. Cellular Accumulation Assay with Hoechst 33342 and Calcein AM

Cellular accumulation assay with hoechst 33342 and calcein AM was performed as described previously [26,27]. Briefly, MDCKII-par, MDCKII-ABCB1, MDCKII-ABCG2, and MDCKII-ABCC1 cells were seeded at densities of 5.0 × 10^4^, 5.0 × 10^4^, 5.5 × 10^4^, and 6.0 × 10^4^ cells/well, respectively, on a transparent 96-well plate and grown to full confluence for 24 h under standard conditions. The cells were then washed twice with 1 × PBS, after which was added either a dilution of ensartinib in Opti-MEM (1, 5, 10, 25, or 50 µM) or a specific inhibitor to act as a positive control (1 µM LY335979, 1 µM Ko143, or 25 µM MK-571 for ABCB1, ABCG2, or ABCC1, respectively). Cells were pre-incubated for 10 min, then 8 µM hoechst 33342 (a model substrate for ABCB1 and ABCG2) or 2 µM calcein AM (ABCC1 substrate) diluted in Opti-MEM was quickly added to all wells other than those containing blank samples. The intracellular levels of the substrates were then determined at 1 min intervals for 30 min using a microplate reader (Infinite M200 Pro, Tecan, Männedorf, Switzerland). Fluorescence was monitored in bottom mode using excitation/emission wavelengths of 350/465 and 485/535 nm for hoechst 33342 and calcein AM, respectively.

### 4.4. Cellular Accumulation Assay with Daunorubicin and Mitoxantrone

The flow-cytometric inhibitory assay with daunorubicin and mitoxantrone was performed with minor modifications as described previously [26,27]. MDCKII-par, MDCKII-ABCB1, MDCKII-ABCG2, and MDCKII-ABCC1 cell lines were seeded at densities of 22.0 × 10^4^, 15.0 × 10^4^, 25.0 × 10^4^, and 22.0 × 10^4^ cells/well, respectively, on a 12-well plate and then incubated for 24 h to reach approximately 70–80% confluence. Further steps including cell washing, the preparation of drug/model inhibitor dilutions, and their addition to cells were performed as described above. After the 10 min pre-incubation period, 2 µM daunorubicin (a substrate of ABCB1/ABCC1) or 1 µM mitoxantrone (ABCG2 substrate) were promptly added to all wells other than those containing background samples, then the plate was incubated for 1 h under standard conditions. The plates were subsequently placed on ice, the cells were washed twice with ice-cold 1× PBS, and 10× trypsin without phenol red was added. After complete trypsinization, the cells were vigorously resuspended in ice-cold 1× PBS containing 2% FBS. Samples were then transferred into test-tubes and kept on ice until their fluorescence could be measured using a flow cytometer (BD FACSCanto II, Allschwil, Switzerland). The excitation/emission wavelengths for daunorubicin and mitoxantrone were 490/565 and 640/670 nm, respectively.

### 4.5. Inhibitory Assay for Human Recombinant CYP Isoforms

CYP inhibitory assay was performed as described previously [26,27]. Experiments were performed with commercial Vivid CYP screening kits containing insect microsomal fractions enriched with a specific human CYP isoform, human CYP reductase, and in some cases, human cytochrome b5. The potential inhibitory activity of ensartinib toward CYP1A2, CYP2B6, CYP2C8, CYP2C9, CYP2C19, CYP2D6, CYP3A4, and CYP3A5 (i.e., all CYP isoforms that drug regulatory authorities recommend for testing of inhibition by new molecular entities [11,12]) was then examined. Experiments were performed using a kinetic mode in black 96-well plates according to the manufacturer´s instructions. Briefly, 5-point serial dilutions of ensartinib or the model inhibitor recommended by the manufacturer in buffer were transferred into the wells, then a master mix consisting of a CYP isoenzyme with an NADPH regeneration system in buffer was added. After a 10 min pre-incubation, the reaction was initiated by adding a mixture of NADP^+^ and the appropriate Vivid substrate, after which fluorescence was measured at 1 min intervals for 60 min using an Infinite M200 Pro microplate reader (Tecan, Männedorf, Switzerland). The enzyme concentration and incubation interval used for data evaluation (15 min) placed the systems within the linear regions of the appropriate reaction velocity curves. Activity changes due to DMSO were eliminated by pre-dilution of ensartinib in DMSO so that 0.5% DMSO was present at all concentration points. 0.5% DMSO was also present in all controls (i.e., 0 and 100% activity controls, and controls using model inhibitors).

### 4.6. Inhibition of CYP3A4 in Intact HepG2-CYP3A4 Cells

A cell-based method using the P450-Glo CYP3A4 Assay and Screening System with Luciferin-IPA together with the CellTiter-Glo Luminescent Cell Viability Assay was used to measure CYP3A4 inhibition in intact HepG2-CYP3A4 cells [27]. HepG2-CYP3A4 cells were seeded on a transparent 96-well plate at a density of 8.0 × 10^4^ cells/well and cultured for 24 h. The cells were then washed once with 1× PBS and treated with Opti-MEM solutions of ensartinib (5, 10, 15, and 25 µM) or a model inhibitor (10 µM ketoconazole). After a 10 min pre-incubation under standard conditions, the CYP3A4 substrate luciferin-IPA was quickly added to all cells other than background samples at a final concentration of 2 µM and the plates were then incubated for 45 min at room temperature. After incubation, the plates were placed on ice and some of the culture media from the cells was transferred to an opaque white 96-well plate and mixed with Luciferin Detection Reagent in a ratio of 1:1 (v/v). The resulting solution was incubated for 20 min at room temperature, then the samples’ luminescence was measured using a microplate reader (Infinite M200 Pro, Tecan, Männedorf, Switzerland) with an integration time of 250 ms. The remaining media was separated from the cells, then 35 µL of pure Opti-MEM was added and the resulting mixture was incubated for 30 min at room temperature. Afterward, 25 µL of CellTiter-Glo Reagent was added to the wells and the cells were left to lyse for 2 min on an orbital shaker. The resulting lysate was incubated for 10 min at room temperature, after which 50 µL was transferred to an opaque white 96-well plate. Luminescence (which correlates with cell viability) was monitored under conditions identical to those used in the previous step. Metabolic data were normalized to viability values to compensate for possible inaccuracies due to non-uniformity in cell growth or false-positive results resulting from drug cytotoxicity. Activity changes due to DMSO were corrected for by using appropriate vehicle controls (0.05, 0.1, 0.15, and 0.25% DMSO).

### 4.7. Molecular Docking Simulations

In silico molecular docking was performed with minor modifications according to the procedure described recently [26]. The structure of ensartinib was obtained from the Zinc Database [47] and CS ChemOffice version 18.0 (Cambridge Soft, Cambridge, MA, USA) was used to minimize its energy. Structures of ABCB1 were downloaded from the Protein Data Bank ([48]; PDB IDs 4M2S and 6C0V) [49,50]. The Swiss-Model Workspace, accessed via the ExPaSy web server [51], was used to create a model of the inward-facing form of ABCB1 based on the crystal structure of mouse abcb1 (PDB ID 4M2S) [50] and the primary sequence of human ABCB1 (P08183). Swiss-Model Workspace was also used to generate a homology model of the outward-facing form with Gln-556 and Gln-1201 mutated to the catalytically active residues Glu-556 and Glu-1201 using the same human ABCB1 sequence and PDB ID 6C0V [49] as templates. Ensartinib and ABCB1 were prepared for docking using MGL Tools 1.5.6 [52]. Proteins were prepared for docking by removing water, ATP, and ligands, and adding hydrogens and Gasteiger charges. Flexible docking calculations were performed with AutoDock Vina 1.1.2 [53], using a 35 × 35 × 35 grid box positioned over the M-site (x = 18.62, y = 55.05, z = −0.82; flexible residues: Phe-303, Tyr-307, Ile-340, Phe-343, Gln-347, Gln-725, Phe-728, Phe-983, Gln-990), R-site (x = 9.43, y = 83.03, z = 17.98; flexible residues: Thr-240, Asp-241, Leu-244, Lys-826, Phe-994), and H-site (x = 36.45, y = 59.64, z = 16.22; flexible residues: Gln-132, Val-133, Trp-136, Cys-137, Asn-183, Phe-194, Leu-879, Phe-938, Phe-942) of the inward-facing form of ABCB1. For docking into the outward-facing form of ABCB1, the 35 x 35 x 35 grid box was positioned over the nucleotide binding domains (NBDs) (x = 172.27, y = 190.31, z = 132.08; flexible residues: Asp-164, Tyr-401, Arg-404, Ile-409, Lys-433, Thr-435, Gln-475, Gln-1175, Gln-1180, and x = 156.69, y = 168.89, z = 119.33; flexible residues: Gln-530, Tyr-1044, Arg-1047, Val-1052, Lys-1076, Gln-1118, His-1232). The exhaustiveness parameter was set to 8 for all docking calculations. Docking results were visualized with PyMOL 1.8.6.0 (The PyMOL Molecular Graphics System, Schrödinger, LLC).

ABCG2 structures (PDB IDs 6HIJ, 6HBU) [54,55] were downloaded from the Protein Data Bank and prepared for docking using MGL Tools 1.5.6 in the same manner as described for ABCB1. The crystal structure deposited into the Protein Data Bank under PDB ID 6HBU features the E211Q mutation. Swiss-Model Workspace was therefore used to change Gln-211 in the NBDs into Glu-211 using the primary sequence of human ABCG2 (Q9UNQ0) and the 6HBU crystal structure as templates. AutoDock Vina was then used to dock ensartinib into the ligand-binding internal cavity of the modified ABCG2 structure (x = 129.81, y = 129.91, z = 142.89, PDB ID 6HIJ; flexible residues: Phe-432, Phe-439, Leu-539, Ile-543, Val-546, Met-549) and NBDs (x = 113.32, y = 92.03, z = 129.89, and x = 94.09, y = 115.37, z = 129.96; PDB ID 6HBU; flexible residues: Thr-82, Lys-86, Gln-126, Glu-211, His-243). The size of the grid box was 35 × 35 × 35 and the exhaustiveness parameter was set to 8 for both docking calculations.

The crystal structure of CYP3A4 containing ketoconazole was downloaded from the Protein Data Bank (PDB ID 2V0M) [56]. Water molecules and ketoconazole were removed, and hydrogens and Gasteiger charges were added using MGL Tools 1.5.6. The heme complex was left in the rigid part of the protein backbone. Docking was again performed with Autodock Vina 1.1.2 (x = 46.90, y = −33.72, z = 37.02; flexible residues: Leu-210, Ile-301, Phe-241, Leu-482, Arg-372) using the same grid box size and exhaustiveness as for the transporters.

### 4.8. ABCB1 ATPase Assay

The ABCB1 ATPase assay was performed with minor modifications as described previously [24]. The ATPase activitity was determined using ABCB1 PREDEASY ATPase Kit according to the manufacturer’s instructions. Sf9 cell membranes (4 µg protein per well) were mixed with various concentrations of ensartinib. The reaction mixture was then incubated for 10 min at 37 °C in the presence or absence of 1.2 mM sodium orthovanadate. The ATPase reactions were started by the addition of 10 mM ATP magnesium salt and stopped 10 min later. After 30 min incubation, absorbance was measured at 590 nm using the microplate reader (Infinite M200, Tecan, Salzburg, Austria). Phosphate standards were present in each plate; verapamil was used as positive control for ABCB1 stimulation.

### 4.9. MTT Proliferation Assay

MTT proliferation assay was performed as described previously [27]. Cells were seeded on a 96-well culture plate at densities (in units of cells/well) of 1.3 × 10^4^ for MDCKII-par, MDCKII-ABCB1, MDCKII-ABCG2, and MDCKII-ABCC1; 1.2 × 10^4^ for A431-par, A431-ABCB1, A431-ABCG2 and A431-ABCC1; 1.2 × 10^4^ for NCI-H1299; 2.0 × 10^4^ for Caco-2; 5.0 × 10^4^ for LS174T; and 6.0 × 10^4^ for HepG2-CYP3A4 and HepG2-EV. After seeding, the cells were cultured for 24 h, then the media was removed and the cells were treated with serial dilutions of a drug or drug combination for 48 h. Vehicles containing media and 40% DMSO in media served as 100% and 0% viability controls, respectively. Upon completion of the 48 h period, the cells were washed once with 1× PBS, the media was replaced with an MTT solution in Opti-MEM (1 mg/mL), and the cells were incubated under standard conditions for 60 min. MTT solution was carefully removed and the resulting formazan crystals were immediately solubilized in DMSO. The cells were then shaken gently for 10 min, after which absorbance of their lysate was measured at 570 nm and 690 nm using a microplate reader (Infinite M200, Tecan, Salzburg, Austria); background values obtained at 690 nm were subtracted from the measurements obtained at 570 nm. This method was used to assess the effect of ABC transporters on sensitivity to ensartinib, in drug combination assays, and to identify drug concentrations appropriate for use in induction studies.

### 4.10. XTT Proliferation Assay

The MTT method is not readily adapted for use with suspension cell cultures, therefore the XTT assay was used to assess the proliferation of HL60 cells. Briefly, HL60-par, HL60-ABCB1, HL60-ABCG2, and HL60-ABCC1 cells were seeded at 2 × 10^4^ cells/well in 96-well culture plates. Immediately after seeding, dilutions of the tested drugs or drug combinations were added to the cells. The plates were then incubated under standard conditions (37 °C, 5% CO_2_) for 48 h, after which XTT solution in PBS (1 mg/mL) and phenazine methosulfate (7.50 µg/mL) were added to the cells and the plates were incubated for another 5 h. Finally, the cells’ absorbance was measured at 450 nm using a microplate reader (Infinite M200, Tecan, Salzburg, Austria). This method was used to assess the effect of ABC transporters on sensitivity to ensartinib and in drug combination assays.

### 4.11. Drug Combination Assays

Experiments focusing on the modulation of ABC transporter-mediated resistance were performed as described previously [26,27]. The parent MDCKII and HL60 cells and their ABCB1-, ABCG2-, and ABCC1-overexpressing counterparts were seeded at the densities specified above onto 96-well culture plates and cultured for 24 h. The medium was subsequently replaced with serial dilutions of ensartinib alone or with various concentrations of a model MDR-susceptible cytostatic agent (daunorubicin or mitoxantrone) with or without 10 µM ensartinib. After 48 h incubation, the cells’ viability was determined as described above. In addition to performing IC_50_-shift analyses, the viability measurements were converted to FA (fraction of cells affected) values and the effects of drug combinations were quantified using the Chou-Talalay combination index (CI) method as implemented in CompuSyn 3.0.1 (ComboSyn Inc., Paramus, NJ, USA). The effects of concomitantly administered drugs were classified as synergistic (CI < 0.9), additive (0.9 < CI < 1.1), or antagonistic (CI > 1.1) based on the computed CI values [57].

In enzyme-oriented studies, 24 h after seeding plates with HepG2-CYP3A4 and HepG2-EV cells, the media was replaced with fresh media containing ketoconazole (10 µM), ensartinib (15 or 25 µM), or 1 µM docetaxel, either individually or in various combinations. Cell viability was measured after a 48 h incubation using the MTT method as described above. Since (1) ketoconazole and ensartinib brought also some level of their own cytotoxicities into the combination effects and (2) Chou-Talalay method is not applicable in this case, we therefore evaluated the sensitization effect of the drug combinations using the following equation:
*sensitization effect* = (*Vd* − *Vdke*) − (100 − *Vke*)
where, *Vd* is the viability for docetaxel alone, *Vdke* is the viability for docetaxel in combination with ketoconazole or ensartinib, and *Vke* is the viability for ketoconazole or ensartinib alone. Sensitization effects of < 0%, 0%, and >0% indicate antagonism, additivity, and synergism, respectively.

### 4.12. MDCKII Monolayer Transport Assay

Transport assays were performed on microporous polycarbonate membrane filters (3 µm pore size, 24 mm diameter, Transwell 3414; Costar, Corning, NY, USA) using MDCKII parental as well as ABCB1-, ABCG2-, and ABCC1-overexpressing MDCKII cells. The cells were seeded at a density of 1.5 × 10^6^ cells per insert and cultured to full confluence for 7 days with media exchanges on the second and fourth days. Before the start of the transport experiments, inserts were washed with 1 × PBS on both the apical and basal sides. The cells were then preincubated in Opti-MEM with or without model inhibitors (1 µM LY335979, 1 µM Ko143, or 25 µM MK-571). The experiments were initiated by replacing the medium with 1 µM ensartinib in Opti-MEM, with or without model inhibitors in the donor chambers. Samples were then collected from the acceptor chambers after 0.5, 1, 2, and 4 h. Ensartinib levels in samples were measured by UHPLC-MS/MS (see below). At the end of the experiment, the integrity of the cellular monolayers was verified using fluorescein isothiocyanate-labeled dextran (MW = 40 kDa); leakage levels up to 5% per hour were considered acceptable.

### 4.13. UHPLC-MS/MS Analysis

Samples from the monolayer transport assays (10 µL) were mixed with 100 µL of acetonitrile and 100 µL of methanol, vortexed for 15 min, and centrifuged at 14,500 rpm for 3 min. The supernatants were then filtered through 0.22 µm PTFE syringe filters into glass vials for analysis. One microliter of the filtered sample was used in the subsequent UHPLC-MS/MS analysis. Detection of ensartinib was performed on an Agilent 1290 Infinity II UHPLC system coupled to an Agilent 6470 QqQ mass spectrometer (Agilent Technologies, Santa Clara, CA, USA). Elution was performed at a steady flow rate of 0.4 mL/min using 0.1% formic acid in water (solvent A) and acetonitrile (solvent B). Elution was performed over 5 minutes using the following eluent composition: 0–0.5 min, 70:30 A:B; 0.5–3.0 min, gradient from 70:30 to 5:95 A:B; 3.0–4.0 min, 5:95 A:B; 4.0–5.0 min, 70:30 A:B. Thermostats were used to maintain the autosampler at 15 °C and the column at 30 °C. The column was a Zorbax Eclipse plus RRHD C18 2.1 × 50 mm, 1.8 µm unit (PN 959757-902). The MS source parameters were set to the following: drying gas, 320 °C, flow rate 10 l/min; sheath gas, 400 °C at 12 l/min; nebulizer pressure 25 psi; capillary voltage 4000 V; nozzle voltage 0 V. Transitions of [M+H]^+^ ions were detected using a dwell time of 150 ms, a cell acceleration of 4 V, and fragmentor voltages of 137 V for 547→357, 257 and 120 transitions (corresponding to collision energies of 16, 32, and 56 V).

### 4.14. Gene Induction Studies

Gene induction studies were performed with minor modifications as described recently [27]. Although all gene induction studies involved the use of quantitative real-time reverse transcription PCR (qRT-PCR) to measure mRNA levels, the methods used for specific cell lines varied considerably. Therefore, they are described separately.

For transporter-oriented studies, A549 (24 × 10^4^ cells/well), NCI-H1299 (18 × 10^4^ cells/well), Caco-2 (50 × 10^4^ cells/well), or LS174T (100 × 10^4^ cells/well) cells were plated in 12-well plates 24 h before the experiment. Subsequently, the medium was replaced with fresh medium containing 0.5 µM ensartinib, 25 µM rifampicin, or 0.1% DMSO (vehicle control). Samples were collected at 24 and 48 h intervals using TRI Reagent, after which total RNA was isolated from the cells using chloroform and isopropanol following the manufacturer’s protocol. The RNA yield was measured using a NanoDrop ND-1000 spectrophotometer (American Laboratory Trading, East Lyme, CT, USA) while RNA quality and integrity were verified by agarose gel electrophoresis. DNase treatment was not performed before reverse transcription because all TaqMan qRT-PCR systems were designed to span introns and/or cross intron/exon boundaries. Using the gb Reverse Transcription Kit, 1000 ng of RNA was transcribed into cDNA. The expression of *ABCB1*, *ABCG2* and *ABCC1* was assessed using gb Easy PCR Master Mix and TaqMan qRT-PCR systems in 384-well plates according to the manufacturer´s instructions, amplifying 20 ng of cDNA per reaction. qRT-PCR was performed using QuantStudio 6 (Life Technologies, Carlsbad, CA, USA) with predefined thermal cycling conditions (95 °C for 3 min then 40 repeats of a cycle of 95 °C for 10 s and 60 °C for 20 s). Relative expression values were determined by comparing each target gene’s expression to the geometric mean expression of the *B2M* and *HPRT1* housekeeping genes using the 2^ΔΔCt^ method.

For enzyme-oriented experiments, the methodology used for A549 and NCI-H1299 cells was as described above. For HepaFH3/upcyte hepatocytes, the procedure described below was followed. Before gene induction studies, viability assays were performed in these hepatocyte models to identify suitable drug concentrations. Briefly, HepaFH3 and upcyte cells (1.25 × 10^4^ cells/well) were seeded on collagen I-coated 96-well plates. After 4 days, the medium was replaced with fresh medium containing different concentrations of ensartinib. The cells were then incubated for 72 h with daily medium replacement and their viability was analyzed with the CellTiter-Glo 2.0 Viability Assay according to the manufacturer´s protocol. Luminescence was measured using a microplate reader (FLUOstar Omega, BMG Labtech, Ortenberg, Germany). For gene induction studies, HepaFH3/upcyte hepatocytes were seeded at 1.6 × 105 cells/well on collagen I-coated 12-well plates 4 days before the experiment to reach full confluence. At time 0, the medium was replaced with fresh medium containing 0.5 µM ensartinib, 50 µM omeprazole, 750 µM phenobarbital, 25 µM rifampicin, or 0.1% DMSO as a vehicle control. Total RNA was isolated after 72 h of incubation with daily medium exchange (with the fresh medium containing the same drug concentrations as the initial medium) using the InnuPREP RNA Mini Kit according to the manufacturer’s instructions. Total RNA content was measured using a NanoDrop instrument and 3 µg of RNA per sample were subjected to DNase I digestion followed by agarose gel electrophoresis to verify RNA integrity. Reverse transcription was conducted using the RevertAid reverse transcriptase and 1 µg of DNase-digested RNA. *CYP1A2* (forward sequence: 5′-TCGTAAACCAGTGGCAGGT-3′; reverse: 5′-GGTCAGGTCGACTTTCACG-3′), *CYP2B6* (forward sequence: 5′-CTCTCCATGACCCACACT-3′; reverse: 5′-CTCTCAGAGGCAGGAAGTTG-3′), and *CYP3A4* (forward sequence: 5′-GTGGGGCTTTTATGATGGTCA-3′; reverse: 5′-GCCTCAGATTTCTCACCAACACA-3′) mRNA levels were determined using Maxima Probe qPCR Master Mix and EvaGreen in 96-well plates according to the manufacturer´s instructions, amplifying 0.3 µL of cDNA per reaction. Relative quantification of the examined CYPs was performed using the 2^−ΔΔCt^ method; the geometric mean of the *GAPDH* (forward sequence: 5′-TGCACCACCAACTGCTTAGC-3′; reverse: 5′-GGCATGGACTGTGGTCATGAG-3′) and *SDHA* (forward sequence: 5′-CGAACGTCTTCAGGTGCTTT-3′; reverse: 5′-AAGAACATCGGAACTGCGAC-3′) levels was used as an internal control to normalize the variability in expression levels. qRT-PCR was performed using a CFX96 Touch Real-Time PCR Detection System (Bio-Rad, Hercules, CA, USA) with initial denaturation at 95 °C for 3 min and 45 repeats of a cycle of 95 °C for 10 s, 62 °C for 10 s, and 72°C for 30 s, followed by melting curve analysis.

### 4.15. Statistical and Data Analysis

All statistical analyses and data evaluations were performed using GraphPad Prism version 8.0.1 (GraphPad Software Inc., La Jolla, CA, USA). *p* values were computed using either one-way ANOVA followed by Dunnett’s post hoc test or the two-tailed unpaired *t*-test as appropriate; differences of *p* < 0.05 were considered statistically significant. Absolute IC_50_ values were determined by non-linear regression using sigmoidal Hill kinetics. Three independent replicates were performed for all experiments, with three biological replicates per experimental replicate.

## 5. Conclusions

In summary, we have shown that ensartinib inhibits several ABC drug efflux transporters and CYP drug metabolizing enzymes, and has the potential to perpetrate clinically relevant DDIs. Additionally, we have demonstrated that ensartinib is an effective antagonist of ABCB1-, ABCG2-, and CYP3A4-mediated cytostatic resistance. Targeting of multiple mechanisms of pharmacokinetic MDR is a particularly desirable characteristic for an MDR modulator. Importantly, ensartinib’s modulatory properties are not compromised by the functional activity of ABC transporters or by interference with their expression, making it a promising candidate for combination anticancer therapy. The results of this complex in vitro study will serve as a valuable foundation for follow-up in vivo testing.

## Figures and Tables

**Figure 1 cancers-12-00813-f001:**
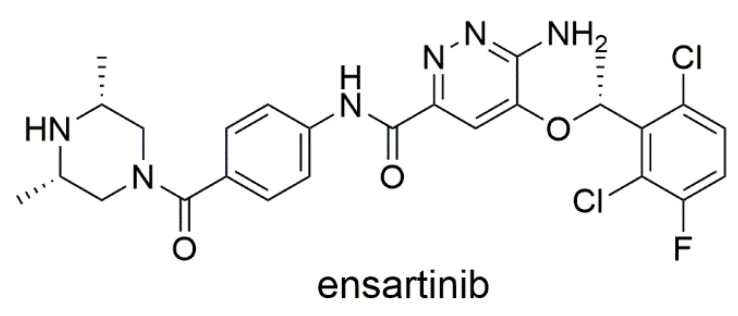
Chemical structure of ensartinib.

**Figure 2 cancers-12-00813-f002:**
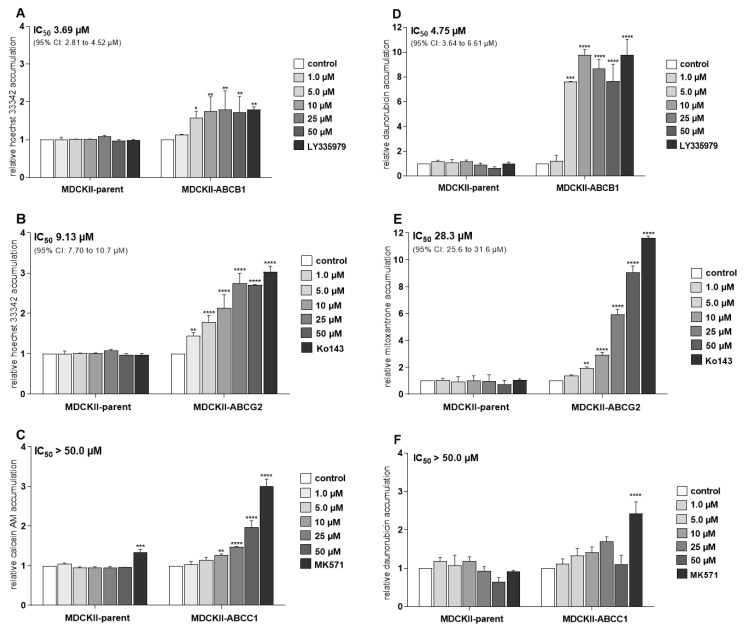
Effects of ensartinib on the intracellular accumulation of hoechst 33342 (**A**,**B**), calcein AM (**C**), daunorubicin (**D**,**F**) and mitoxantrone (**E**) in MDCKII cells transduced with the human ATP-binding cassette (ABC) transporters ABCB1 (**A**,**D**), ABCG2 (**B**,**E**), and ABCC1 (**C**,**F**). In hoechst 3342 and calcein AM assays, cells were pre-incubated with ensartinib at various concentrations for 10 min. The substrates (8 µM hoechst 3342 or 2 µM calcein AM) were then added and their accumulation in intact cells was monitored directly using a fluorimeter in kinetic mode. The endpoint interval was used for data evaluation. In daunorubicin and mitoxantrone assays, cells were pre-incubated with ensartinib or model inhibitors for 10 min, then treated with 2 µM daunorubicin or 1 µM mitoxantrone, respectively. After 1 h incubation, the cells were trypsinized and the substrates’ fluorescence was measured with a flow cytometer. In both cases, LY335979 (1 µM), Ko143 (1 µM), and MK571 (25 µM) were used as model ABCB1, ABCG2, and ABCC1 inhibitors, respectively. These inhibitor concentrations induced maximal inhibition of the transporters, so the results obtained with the model inhibitors were taken to represent 100% inhibition when calculating absolute IC_50_ values for ensartinib. The plotted relative accumulation values represent fold-increase in probe substrate accumulation elicited by tested compound and are expressed as ratios of relative fluorescence units (RFUs) from treated samples to RFUs of control in particular cell subline. The data are means ± SD based on three independent experiments. Statistical analysis was performed using one-way ANOVA followed by Dunnett’s post hoc test (* *p* < 0.05; ** *p* < 0.01; *** *p* < 0.001; **** *p* < 0.0001 relative to control).

**Figure 3 cancers-12-00813-f003:**
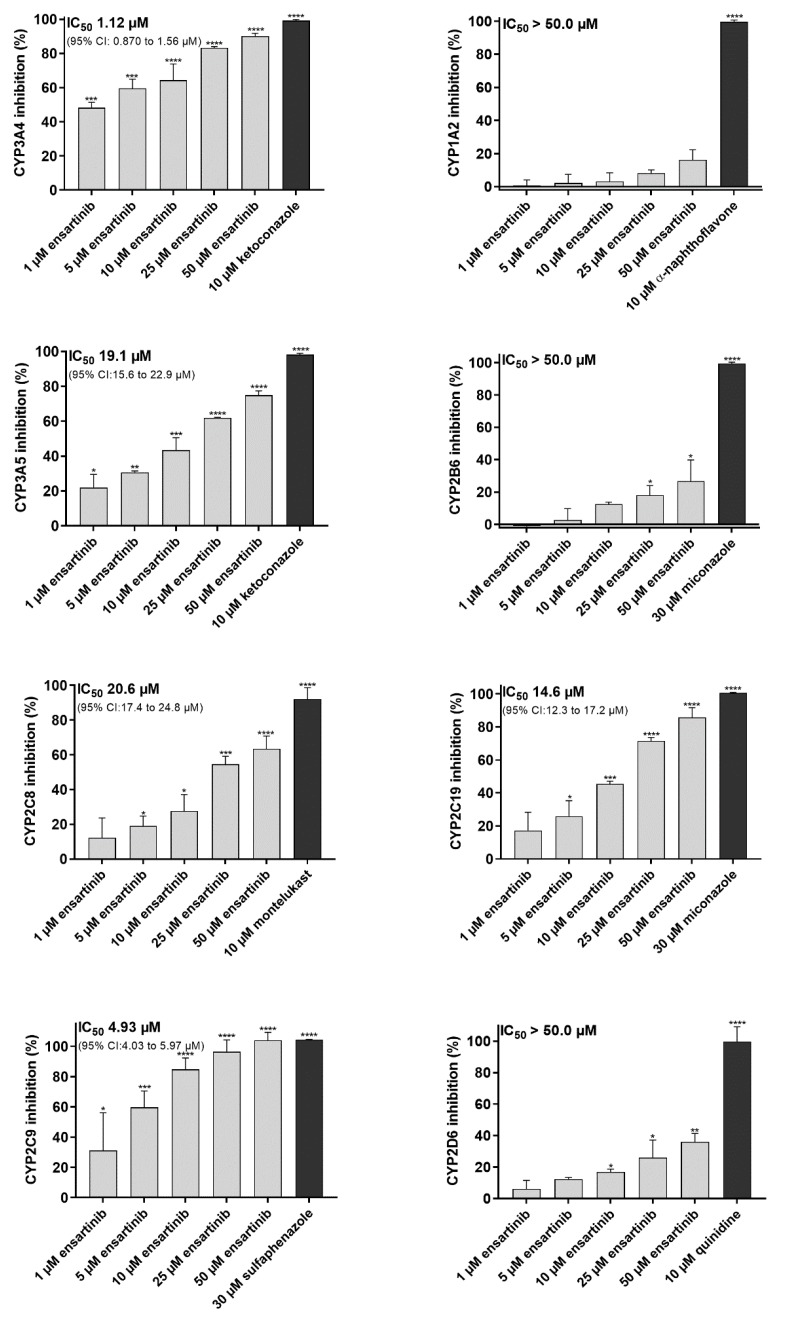
Effect of ensartinib on the activities of human cytochrome P450 (CYP)1A2, CYP3A4, CYP3A5, CYP2B6, CYP2C19, CYP2C8, CYP2C9, and CYP2D6 isoforms assessed using the commercial Vivid CYP Screening Kits. Ensartinib and model inhibitors were pre-incubated with enzymes for 10 min, after which the reaction was initiated by adding a mixture of NADP^+^ and the appropriate Vivid substrate. Raw fluorescence measurements acquired after 15 minutes’ incubation were normalized against reference 100% and 0% activity values. Maximal (100%) enzyme activity was represented by the fluorescence of samples containing only the enzyme and 0.5% DMSO with no drug. The level of fluorescence corresponding to 0% activity was determined by analyzing samples incubated with the enzyme solvent buffer without the enzyme and 0.5% DMSO. Absolute IC_50_ values were determined by assuming that the model inhibitor response represented 100% inhibition. The plotted values are means ± SD based on three independent experiments. Statistical analysis was performed with background-subtracted raw fluorescence data using one-way ANOVA followed by Dunnett’s post hoc test (* *p* < 0.05; ** *p* < 0.01; *** *p* < 0.001; **** *p* < 0.0001 compared to 100% activity control).

**Figure 4 cancers-12-00813-f004:**
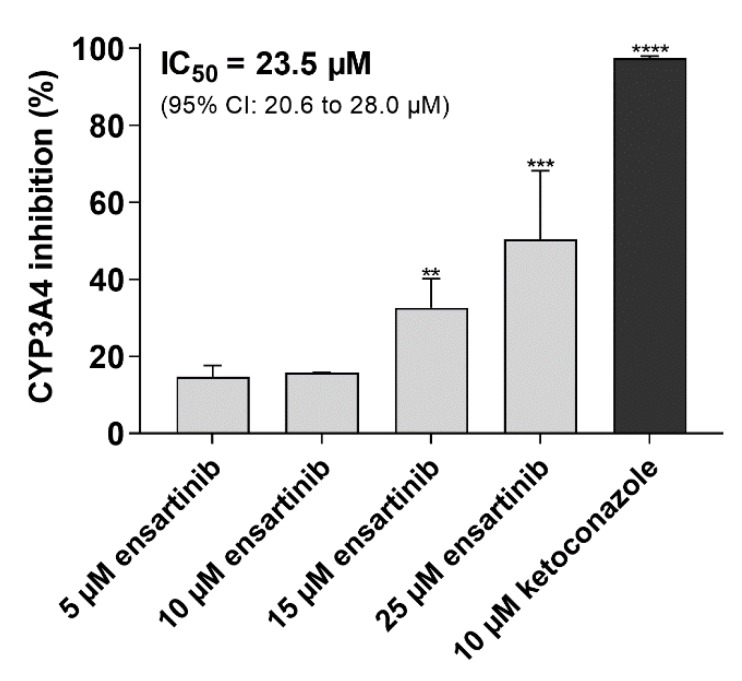
Effect of ensartinib on human CYP3A4-mediated metabolic activity in HepG2-CYP3A4 cells. Cells were pre-incubated with the tested drugs for 10 min, then the reaction was initiated by adding the substrate (luciferin-IPA). The reaction was stopped after 45 min and the samples’ luminescence was measured to assess the cells’ viability. Metabolic data were normalized to viability values to compensate for possible inaccuracies caused by non-uniformity in cell growth and/or false-positive results due to drug cytotoxicity. The viability values were then re-normalized as inhibition percentages. The plotted values are means ± SD based on three independent experiments. Statistical analysis was performed using background-subtracted luminescence data normalized to viability values using one-way ANOVA followed by Dunnett’s post hoc test (** *p* < 0.01; *** *p* < 0.001; **** *p* < 0.0001 compared to respective vehicle control).

**Figure 5 cancers-12-00813-f005:**
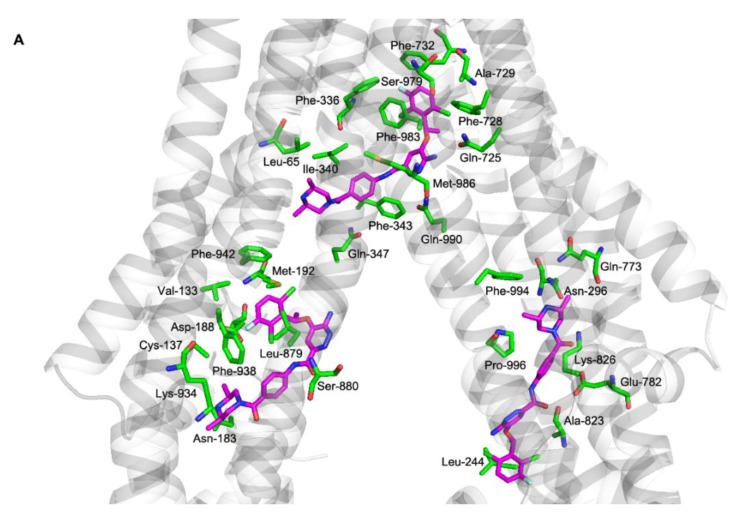
Flexible molecular docking of ensartinib into ABCB1, ABCG2, and CYP3A4 structures and analysis of ensartinib’s interactions with ABCB1 ATPase domain using ATPase assay. (**A**) Molecular docking of ensartinib into the inward-facing form of ABCB1. The top-ranked poses of ensartinib (−12.6, −9.5, and −9.3 kcal/mol) are shown for M-, H-, and R-sites, respectively. Ensartinib is shown in magenta, and protein residues within 4 Å of the ligand are shown as green sticks and labeled. (**B**) Flexible molecular docking of ensartinib into nucleotide binding domains (NBDs) of the outward-facing form of ABCB1. Ensartinib is shown in magenta, and the ATP molecule that originally co-crystallized with the protein backbone is shown using orange sticks. Residues within 4 Å are green, predicted H-bonds are depicted as yellow dashed lines. (**C**) Effects of ensartinib on the vanadate-sensitive ATPase activity of ABCB1-Sf9 membrane fraction. Lower dotted line represents baseline vanadate-sensitive ATPase activity, while upper dotted line shows activated ATPase activity triggered by a reference substrate verapamil. In the inhibition and activation experiments, reductions in the stimulated ATPase activity (indicating inhibitory interaction of the drug toward ABCB1’s ATPase) and increases in baseline ATPase activity (indicating substrate properties of the drug) were recorded, respectively. Statistical analysis of differences between stimulated control and ensartinib-treated samples in inhibition assays (* *p* < 0.05; ** *p* < 0.01) as well as analysis of differences between baseline control and ensartinib-treated samples in activation experiments († *p* < 0.05; †† *p* < 0.01) were determined using two-tailed unpaired *t* test. Data are expressed as means ± SDs from three independent experiments. (**D**) Two possible orientations of ensartinib in ABCG2 identified by flexible docking analysis (−12.9 and −12.5 kcal/mol). ABCG2 protein residues predicted to be within 4 Å of ensartinib are show as green sticks. (**E**) Binding of ensartinib to the CYP3A4 structure. Ensartinib is shown in magenta, heme in yellow, and protein residues surrounding the ligand are depicted as green sticks.

**Figure 6 cancers-12-00813-f006:**
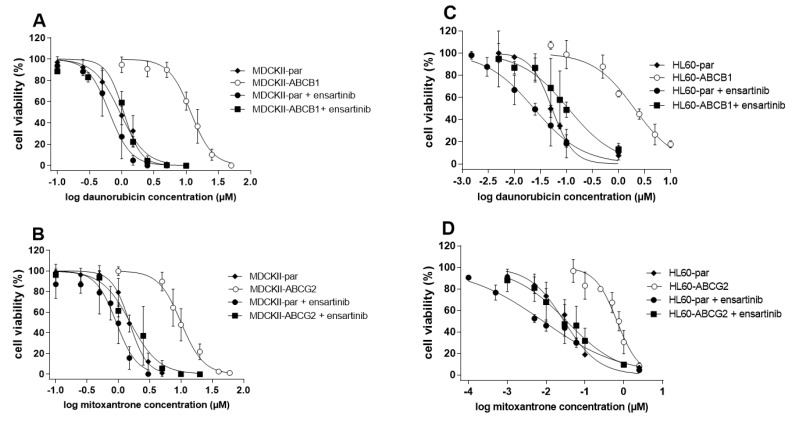
Effect of the combination of 10 µM ensartinib with daunorubicin or mitoxantrone in MDCKII-ABCB1/HL60-ABCB1 (**A**,**C**) or MDCKII-ABCG2/HL60-ABCG2 (**B**,**D**) cells, respectively. Drug-induced effects detected in the parental cell lines are shown for each variant. Cells were exposed to daunorubicin or mitoxantrone either alone or in combination with ensartinib for 48 h, after which cell viability was measured by the MTT method. The presented data are means ± SD for at least three independent experiments.

**Figure 7 cancers-12-00813-f007:**
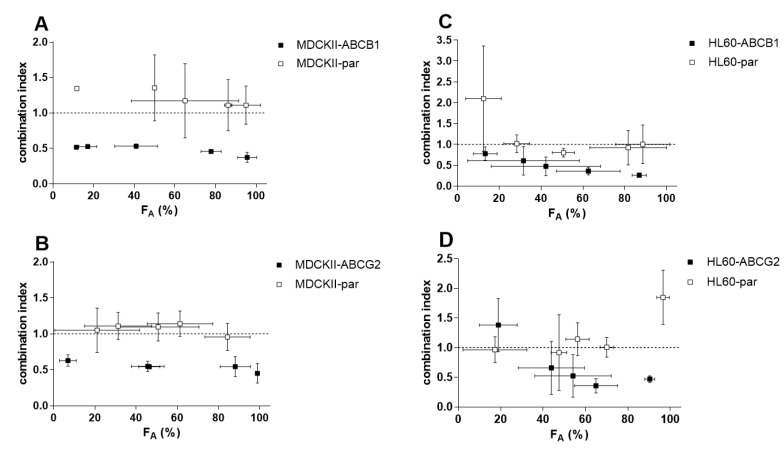
F_A_-combination index (CI) plot for the combinations of 10 µM ensartinib with daunorubicin or mitoxantrone in MDCKII-ABCB1/HL60-ABCB1 (**A**,**C**) and MDCKII-ABCG2/HL60-ABCG2 (**B**,**D**) cells, respectively. The results obtained with the same combinations in the parental cell lines are also shown. F_A_-CI plots were generated by using CompuSyn software to analyze the combination data shown in Figure 6 and ensartinib proliferation data (not shown). Combination outcomes can be synergistic (CI < 0.9), additive (0.9 < CI < 1.1), or antagonistic (CI > 1.1). The plotted data points are means ± SD from at least three independent experiments.

**Figure 8 cancers-12-00813-f008:**
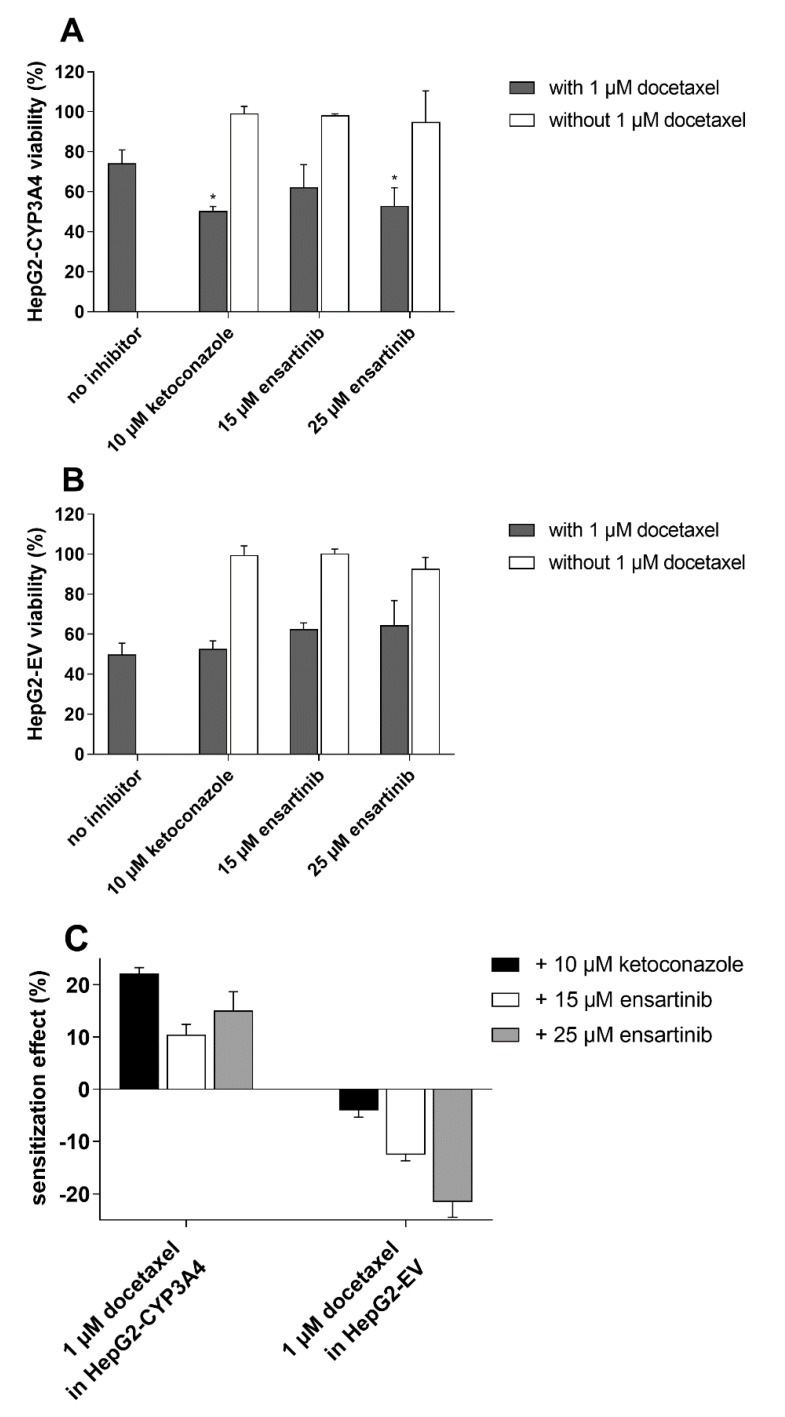
Effects of ensartinib (15 and 25 µM) and the model CYP3A4 inhibitor ketoconazole (10 µM) on the antiproliferative activity of 1 µM docetaxel in HepG2-CYP3A4 (**A**) and HepG2-EV (**B**) cells. Cells were exposed to the tested drugs alone or in tandem for 48 h, after which their viability was assessed using the MTT test. (**C**) Sensitization effects were computed using the data from viability measurements. Statistical analysis was performed using one-way ANOVA followed by Dunnett’s post hoc test (* *p* < 0.05 compared to no inhibitor). The plotted values are means ± SD for three independent experiments.

**Figure 9 cancers-12-00813-f009:**
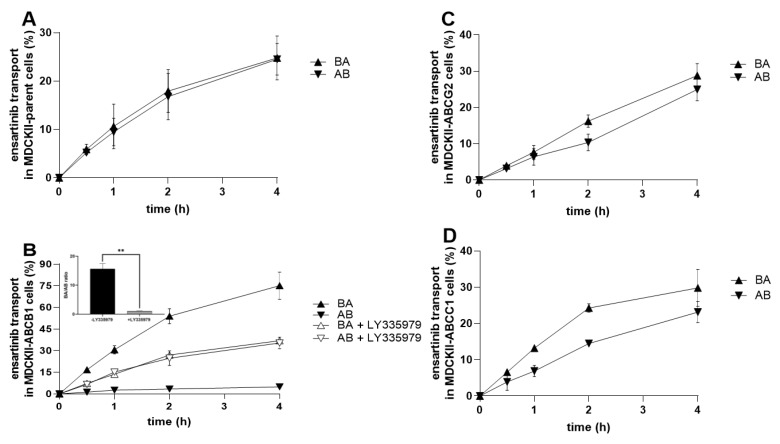
Transport of 1 µM ensartinib across MDCKII-parent (**A**), MDCKII-ABCB1 (**B**), MDCKII-ABCG2 (**C**), and MDCKII-ABCC1 (**D**) cell monolayers. LY335979 (1 µM) was used as a control ABCB1 inhibitor. Cell monolayers were exposed to ensartinib or a combination of ensartinib with a model inhibitor in the donor compartment, and sampling was performed in the acceptor chambers at selected time points. Ensartinib levels were assessed by HPLC-MS/MS. Transport ratios (*r*) were calculated by dividing the percentage of ensartinib transported in the BA direction by that in the AB direction 4 h after drug addition. The statistical significance of changes in *r* ratios in the absence/presence of the model inhibitor was analyzed using the two-tailed unpaired *t*-test (** *p* < 0.01).

**Figure 10 cancers-12-00813-f010:**
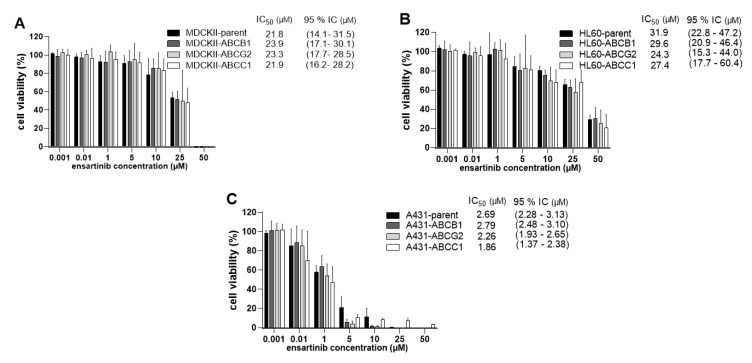
Sensitivity of MDCKII (**A**), HL60 (**B**), and A431 (**C**) sublines to ensartinib. The cells were treated with ensartinib for 48 h and their viabilities were measured using the MTT (MDCKII, A431) or XTT (HL60) proliferation tests. One-way ANOVA followed by Dunnett’s post hoc test was used for statistical comparisons. The viabilities of the parent cells were compared to those of transporter-overexpressing cells for each concentration; no statistically significant differences were observed for any variant. Additionally, the IC_50_ values of transduced and parent cells were compared using the two-tailed unpaired *t*-test. Again, no statistically significant differences were observed. The plotted values are means ± SD based on three independent experiments.

**Figure 11 cancers-12-00813-f011:**
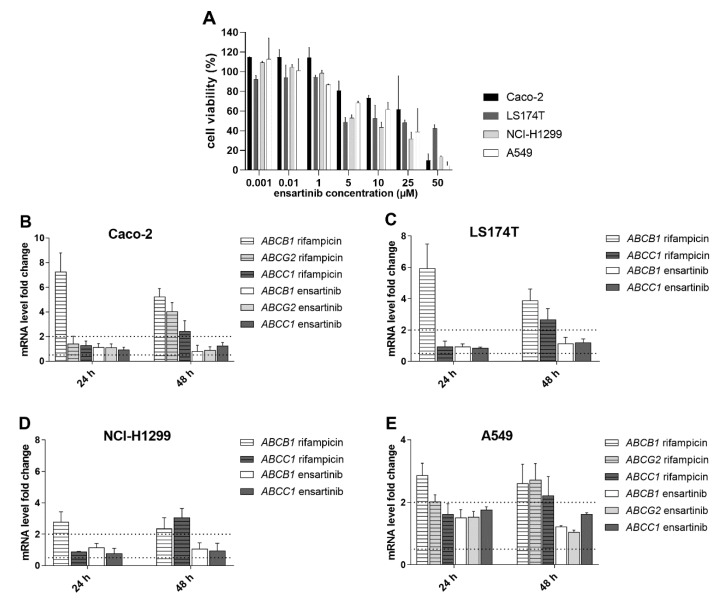
Changes in *ABCB1*, *ABCG2*, or *ABCC1* mRNA levels in physiological Caco-2 and LS174T cells (**B**,**C**) and non-small cell lung cancer (NSCLC) cellular models (**D**,**E**) after treatment with ensartinib. The tested drug concentration was chosen based on the results of MTT viability experiments (**A**) and previously determined Cmax values. During the gene induction assays, the cells were incubated with 0.5 µM ensartinib or 25 µM rifampicin for 24 and 48 h, and the mRNA levels of target genes were evaluated by qRT-PCR. *ABCG2* was not detectable in LS174T and NCI-H1299 cells. The boundaries of downregulation/upregulation positivity based on the EMA’s recommendations are indicated by dotted lines. The plotted data are means ± SD from at least three independent experiments.

**Figure 12 cancers-12-00813-f012:**
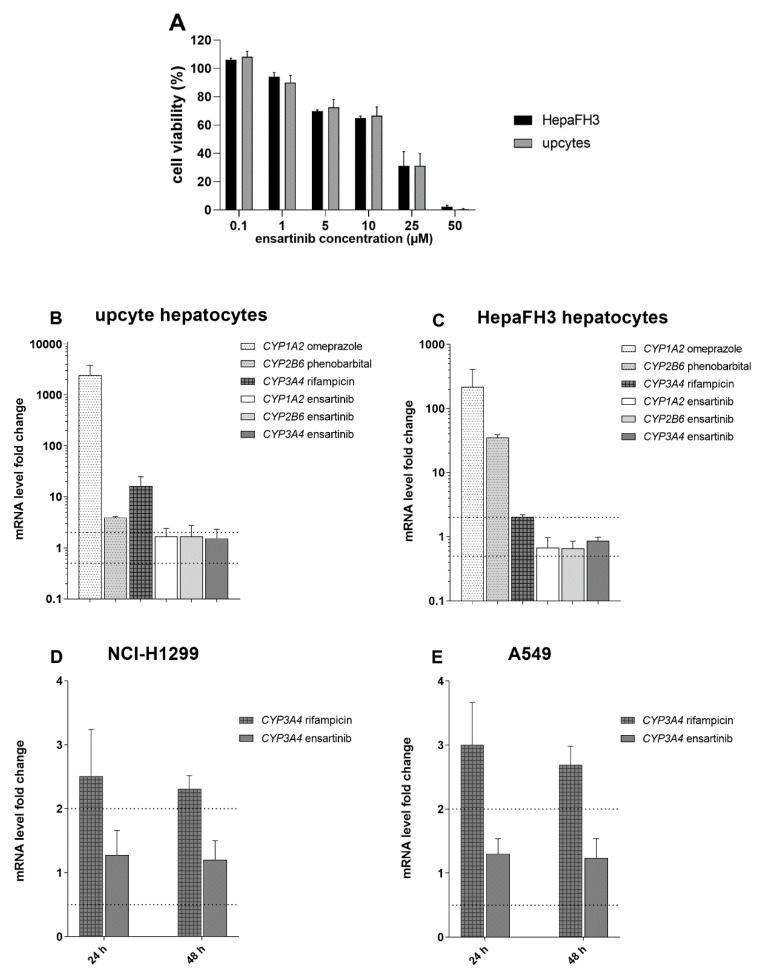
Effect of ensartinib on *CYP1A2*, *CYP2B6* and *CYP3A4* expression. Studies on its potential to perpetrate drug-drug interactions (DDIs) were performed in physiological models, namely upcyte (**B**) and HepaFH3 (**C**) hepatocytes. Assessments of possible changes in MDR phenotype were performed in the NSCLC cellular models NCI-H1299 (**D**) and A549 (**E**). The tested ensartinib concentration (0.5 µM) was chosen based on previously determined Cmax values and cell viability measurements (**A**). For hepatic models, cells were incubated with 0.5 µM ensartinib or with the model inducers omeprazole (50 µM), phenobarbital (750 µM), or rifampicin (25 µM). After 72 h, the mRNA levels of the chosen genes were analyzed by qRT-PCR. The NSCLC cells were incubated with 0.5 µM ensartinib or 25 µM rifampicin for 24 and 48 h, after which the mRNA levels of *CYP3A4* were evaluated by qRT-PCR. The boundaries of downregulation/upregulation positivity based on the EMA’s recommendations are indicated by dotted lines. The plotted data are means ± SD based on at least three independent experiments.

**Table 1 cancers-12-00813-t001:** IC_50_-shift analysis of the combination effects of ensartinib (10 µM) with the cytostatic substrates daunorubicin or mitoxantrone in MDCKII and HL60 cell lines.

Cell Line	Drug(s)	IC_50_ ^1^ (µM)	95% CI (µM)	R_R_^2^
MDCKII-parent				
	daunorubicin	0.947	(0.814–1.07)	
	mitoxantrone	1.59	(1.39–1.64)	
	daunorubicin + ensartinib	0.673 ^ns^	(0.551–0.743)	1.41
	mitoxantrone + ensartinib	0.955 ^ns^	(0.746–1.13)	1.66
MDCKII-ABCB1				
	daunorubicin	11.9	(10.7–12.9)	
	daunorubicin + ensartinib	1.06 ^****^	(0.743–1.22)	11.2
MDCKII-ABCG2				
	mitoxantrone	9.98	(8.91–11.5)	
	mitoxantrone + ensartinib	1.62 ^***^	(1.22–1.98)	6.16
HL60-parent				
	daunorubicin	0.0531	(0.0473–0.0589)	
	mitoxantrone	0.0305	(0.230–0.0370)	
	daunorubicin + ensartinib	0.0221 ^ns^	(0.0169–0.0340)	2.40
	mitoxantrone + ensartinib	0.0089 ^ns^	(0.0059–0.0119)	3.43
HL60-ABCB1				
	daunorubicin	2.39	(1.61–2.53)	
	daunorubicin + ensartinib	0.108 ^****^	(0.0664–0.2248)	22.1
HL60-ABCG2				
	mitoxantrone	0.666	(0.520–0.823)	
	mitoxantrone + ensartinib	0.0359 ^****^	(0.0197–0.0603)	18.6

^1^ These values were calculated using the data shown in Figure 6. The IC_50_ values for the combinations were compared to those of daunorubicin or mitoxantrone alone in the appropriate cell lines using the two-tailed unpaired *t*-test (*** *p* < 0.001; **** *p* < 0.0001); ^2^ The reversal ratio (R_R_) is the ratio of the IC_50_ for single drug treatment to that for the combined drug treatment in the cell line of interest.

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
