# Peer review of "Ensartinib (X-396) Effectively Modulates Pharmacokinetic Resistance Mediated by ABCB1 and ABCG2 Drug Efflux Transporters and CYP3A4 Biotransformation Enzyme"

_cancers, 2020, doi:10.3390/cancers12040813_

Round 1

Reviewer 1 Report

I agree with the authors' responses, however, I still have one concern about this study. In the reversal study, the authors found that 10 ?M ensartinib effectively reversed ABCB1- and ABCG2-mediated drug resistance in HL60-ABCB1 and HL60-ABCG2 cells (Table 1). But in the cytotoxicity assay, 10 ?M ensartinib seemed to be toxic in HL60-ABCG2 cells (cell viability ~70%). Thus it is possible that the decreased IC50 was partially caused by toxicity of ensartinib. The authors should provide more data for validation.

Reviewer 2 Report

The authors have satisfactorily responded to most of my previous concerns, except for the concern #2. I am still curious whether the effect of ensartinib is unique. The authors should at least perform some key experiments to compare the effect of ensartinib with other ALK inhibitors (such as crizotinib, alectinib, ceritinib) and TKIs with different molecular targets (such gefitinib, sorafenib, and etc.).

Round 2

Reviewer 2 Report

The authors have adequately responded to my concern.

This manuscript is a resubmission of an earlier submission. The following is a list of the peer review reports and author responses from that submission.

Round 1

Reviewer 1 Report

The manuscript entitled "Ensartinib (X-396) effectively modulates pharmacokinetic resistance mediated by ABCB1 and ABCG2 drug efflux transporters and CYP3A4 biotransformation enzyme" described the finding of ensartinib modulating the function of ABCB1, ABCG2 and CYP3A4. Overall, the study is interesting. However, there are many concerns that should be clarified as below:

What's the point of mentioning NSCLC in the first paragraph of i-Introduction since the authors didn't focus on human lung cancer cell lines in this study. If the authors want to focus on lung cancer, they should avoid switching between different types of cancer cells. In this study, the authors used HepG2 (liver cancer), A549 (lung cancer), NCI-H1299 (lung cancer), Caco-2 (colon cancer) and LS174T (colon cancer).

In Fig. 2, In the accumulation assays, the data showed same level of accumulation in parental and ABCB1/ABCG2/ABCC1-transfected cell lines, which is hard to explain. In theory, cells that overexpress ABC transporters should display lower substrate accumulation due to the efflux function, while parental cells should show higher drug accumulation. The authors need to check the data carefully. Also, the authors should provide more evidence that the transfected cells overexpress ABCB1/ABCG2/ABCC1 by Western blotting or other immunoblotting approaches.

In Fig. 3, the authors should show statistical analysis of their data and mark the significance in the graph. Same issue as in Fig. 4.

In the docking result, Fig. 5B and 5C should be clearly labeled. Also, interactions between ensartinib and ATP-binding site should be clearly described. The conclusion getting from docking study that ensartinib competing with ATP is not solid enough, in vitro study is needed to provide direct evidence (e.g. ATPase assay).

In Fig. 8C, error bars are missing. In Fig 10, the authors should display statistical analysis results on the graph.

In section 2.8, the authors used 3 cell lines to measure cell viability, however, they used only 1 cell line in previous section to measure the transport ratio. Cell lines should be consistent through the authors' studies in order to control variables.

In Fig. 11B (24 h ABCB1 rifampcin; 48 h ABCB1 rifampicin), 11C (48 h ABCB1 rifampcin), 11D (24 h ABCB1 rifampicin), 11E (24 h ABCB1 rifampicin; 48 h ABCB1 rifampicin) and 12D (24 h CYP3A4 rifampicin), the SD are around 30% to 50%, which makes not reliable enough to support the authors' conclusion. More data is needed to confirm the trend.

Reviewer 2 Report

This paper seems to be the last one in series of papers attempting to evaluate possible pharmacokinetic drug interactions of various compounds, mostly CDK or TKI inhibitors, with ABC transporters. There are interesting results described an the methods are well described.

Comments:

The inhibition of CYPs were checked using Vivid systems. These were developed and are useful for high throughput assays but not for detailed studies on drug interactions. For analyses of binding of drugs to proteins and attempts to evaluate the processes connected with the thermodynamics of drug interactions, more specific substrates should be used rather than less specific Vivid substrates. To confirm the interactions of compounds studied with a specific CYP form, the HepG2 cells transfected with the respective CYP. This technique was developed by Xuan et al (CBI 2016) without proper credit given. Thorough the paper, the P450 enzymes are incorrectly labeled as CYP450. Individual CYP enzymes are named correctly. The reason is that the abbreviation CYP means CYtochromeP450 so there is no need to add 450 again. People involved in CYP studies obey this recommendation

Reviewer 3 Report

In this manuscript entitled “Ensartinib (X-396) effectively modulates pharmacokinetic resistance mediated by ABCB1 and ABCG2 drug efflux transporters and CYP3A4 biotransformation enzyme”, the authors identified that ensartinib, a tyrosine kinase inhibitor (TKI) for mutated anaplastic lymphoma kinase (ALK), is a potential inhibitor of ABCB1 and ABCG2 drug efflux transporters and CYP3A4 and CYP2C9 drug metabolic enzymes. These effects may contribute the synergistic in vitro anticancer activity of ensartinib combining with chemotherapeutic agents (daunorubicin, mitoxantrone and docetaxel). Overall, this is a study with some potential interest. The study design is straightforward and well presented. However, the conclusion was not supported by in vivo studies that will improve the significance of this study. Other concerns are described below.

The protein expressions of ABC transporters in ABCB1/ABCG2/ABCC1-transduced MDCKII, HL60 and A431 cell lines should be confirmed by Western blotting. Similarly, the CYP3A4 expression in HepG2-CYP3A4 and HepG2-EV cells should also be confirmed. In addition, the effects of ensartinib at the used concentrations (10~25 uM) on the protein expressions of ABC transporters and CYP3A4 should be examined. It is unclear that whether other structurally similar TKIs also exhibit similar effects as ensartinib. The authors should perform some key experiments to compare the effects of other TKIs with ensartinib.
